# Workshop, Cost-Effective and Streamlined Fabrications of Re-Usable World-To-Chip Connectors for Handling Sample of Limited Volume and for Assembling Chip Array

**DOI:** 10.3390/s18124223

**Published:** 2018-12-01

**Authors:** Jiann-Hwa Lue, Yu-Sheng Su, Tai-Chih Kuo

**Affiliations:** 1Department of Optometry, Central Taiwan University of Science and Technology, Taichung City 406, Taiwan; 108362@ctust.edu.tw; 2Department of Computer Science and Engineering, National Taiwan Ocean University, Keelung City 202, Taiwan; 3Department of Biochemistry and Molecular Cell Biology, Taipei Medical University, Taipei City 110, Taiwan

**Keywords:** port, world-to-chip connector, double adhesive film, mini grinder, coverslip

## Abstract

The world-to-chip interface is an essential yet intriguing part of making and employing microfluidic devices. A user-friendly connector could be expensive or difficult to make. We fabricated two ports of microfluidic chips with easily available materials including Teflon blocks, double adhesive films, coverslips, and transparency films. By using a mini grinder, coverslips were drilled to form small holes for the fluid passages between port and chip. Except for the double adhesive films, the resultant ports are durable and re-useable. The DK1 port, contains a mini three-way switch which allows users to handle fluid by a tube-connected pump, or by a manual pipette for the sample of trace amount. The other port, the DK2 port, provides secured tube-connections. Importantly, we invented a bridge made of craft cutter-treated transparency films and double adhesive films to mediate liquid flow between DK2 port and chip. With the use of a bridge, users do not need to design new ports for new chips. Also, individual chips could be linked by a bridge to form a chip array. We successfully applied DK1 port on a microfluidic chip where green fluorescent protein was immobilized. We used DK2 port on an array of fish chips where the embryos of zebra fish developed.

## 1. Introduction

Microfluidic devices are recognized as an attractive and powerful tool in modern sciences and technologies [1,2,3,4]. However, the uses of microfluidic devices remain limited. The difficulty of accessing the equipment and materials of (soft) lithography has been partially relieved with the interesting uses of paper, craft cutter, double-sided adhesive films, nylons etc. [5,6,7,8,9,10,11,12,13]. The world-to-chip [14] connectivity is one remaining issue which has to be improved to popularize the use of microfluidic devices.

The fabrications and uses of world-to-chip interconnection have attracted the attention of many researchers (see [15,16] for review). They developed various interconnections with one or several of the features of low cost [17], easy to use [18,19,20,21,22,23], re-usability [20,23,24,25,26,27,28], low dead volume [17], etc. In spite of those reports, many microfluidic devices remain equipped with inconvenient tube connections for example, singularly planted tubes onto individual ports on a sophisticated microfluidic chip, which could be expensive and non-reusable. Drilling holes and attaching ports and tubes to the chip is laborious. Ideally, the chip should be equipped with low-cost and reusable ports which support ‘plug and play’ tube connections. Also, preferably, the port should allow pipette-loading of samples with limited volume. In that case, the use of syringe and tubing is not possible for having large dead volume. Here, we report a workshop style manufacturing process of new world-to-chip interconnections via port modules with many interesting features. Since the port is an integral part of the microfluidic device, the design of port and chip must be considered together. Due to the lack of soft lithography equipment, we made microfluidic chips by placing parallel double adhesive film between a glass slide and coverslip (or transparency film). In fact, cell biologists have used double adhesive tapes to construct sample chambers to observe microorganisms under a microscope [29]. Such primitive design consists an inborn difficulties of handling samples and reagents through the narrow openings between the coverslip and slide. To solve this problem, we developed a gentle and quick way of drilling holes on glass coverslips. The consequence is that port modules can be placed on the holes which lead to channels of the chip. We also used double adhesive films to bind the port and chip. Since the binding is not permanent, the ports can be removed and re-used. With the above basic designs, we added other desirable features in the port modules of microfluidic chips for immobilizing protein and for studying the development of fish embryos. We demonstrate the fabrications and uses of two microfluidic chips. In the first example, green fluorescence protein (GFP) was immobilized in specified area of the protein chip. In the second one, the development of fish embryos was monitored in a configurable microfluidic-chip.

## 2. Materials and Methods

### 2.1. Materials

Sodium phosphates (mono and dibasic, Reagent Plus > 99%, Sigma-Aldrich Inc., St Louis, MO, USA), sodium chloride (>99%, Sigma-Aldrich Inc.), amino-propyl-triethoxy-silane (>98%, Sigma-Aldrich Inc.), glutaraldehyde (25%, Sigma-Aldrich Inc.), and imidazole (>99%, ACS reagent, Sigma-Aldrich Inc.) were purchased through Uni-Onward Inc. (Taipei, Taiwan). Double adhesive films (M060) were kindly provided by Taiwan Super Stick Tech Materials Co. Ltd. (New Taipei, Taiwan). The M60 double adhesive film can resist a pulling force up to 1.2 kg/inch^2^. Microscope slides (“1 × 3”, (1 inch by 3 inch), 1 mm, FEA Taiwan) and coverslips (Deckgläser 24 × 60 mm, Vetrini Coprioggetto) were purchased through Bioman Inc. (New Taipei, Taiwan). Transparency films for projector/printer were bought from a local stationery store. Tygon tubing (inner diameter 1/32″, outer diameter 3/32″, S-50-HL, Saint-Gobain Performance Plastics Co, Solon, OH, USA) was purchased from Dogger Instruments Co. Ltd., Taipei, Taiwan. Recombinant eGFP protein (MW 26.9 kD) was produced by *Escherichia coli* cells and purified (20~40 μM, in 300 mM imidazole, 150 mM NaCl, 10% glycerol, 10 mM sodium phosphate pH 7.4) from bacterial extract using Ni-NTA resins. Craft Robo Lite (GraphTech Inc., Totsuka-ku Yokohama, Japan) and mini grinder (S-Turbo, Octopus Inc., Taipei, Taiwan) were purchased from local sale representatives.

### 2.2. Methods

**Treatment of coverslips with amino silane.** We followed the processes reported previously by us [30]. The coverslips were dried with nitrogen gas and then treated with UV irradiation for 15 min for each side. The cleanness of the coverslips was checked by dropping 3 μL water droplets on the coverslip. The water droplet should be stretched flat over the coverslip. The coverslips were rinsed with double distilled water and dried with high-pressure nitrogen gas. The coverslips were vertically dipped in a container with a 2.5-cm height of 2% APTES (in anhydrous acetone) for 2 min. Note that only the lower end of the coverslip was treated with APTES. The reason for not dipping the coverslips completely into the APTES solution was to avoid proteins being immobilized on the windows of bubble traps. The coverslips were sequentially rinsed with 95% ethanol and dd water and dried with high-pressure nitrogen.

**Fabrication of DK ports.** The DK1 port (Figure 1) was cut from a Teflon block. A 2/64″ drill bit was used to drill tunnels, and a 5/64″ flat-head bit was used to make 3-mm depth craters as the plugs of Tygon tubes. The Teflon block was also drilled with a 5.1 mm flat-head bit to form a cylindrical crater (8-mm depth) for housing a three-way switch (Figure 1b). The crater was further drilled with a 4-mm bit to make a through observation window (Figure 1c) for ensuring the three-way switch being completely pushed into the crater. To make the three-way switch, a short Teflon round stick (5.2 mm diameter, 18 mm length) was drilled to have T-shaped tunnels. Care was taken to ensure that the positions of the holes of T-tunnels matched those in the port’s main body.

The DK2 port was made by installing four tube connectors into a Teflon block (15 × 65 × 10 mm). The Teflon block was sent to a machine shop to have four screw holes drilled. The connectors were taken from the spare tubing nipple parts (code # 19-0035-01, Pharmacia LKB Biotechnology) of a four-way valve (LV-4). Since the resultant holes might be slightly larger than the connectors, the connector threads were wrapped with plumber’s tape. The volume of the channel within the gate of DK2 port is ~28 μL.

**Drilling holes on coverslips.** The power of the mini grinder was tuned to the lowest speed, and with a firm grip of the grinder, the burr-tip was gently lowered onto the coverslip. In a few seconds, a through hole could be made. After several uses, the repeated cracks of coverslips indicate a wear out of the burr-tip which should be replaced. To facilitate the task, a guide plate can be used (see Appendix A).

**Cutting double adhesive and transparency films.** For the lab-on-chip immobilization of GFP protein, the channels and holes in the individual layer of double adhesive films and transparency films were designed by using Adobe Illustrator. A computer-controlled craft cutter (Craft Robo Lite, GraphTech) cuts media according to the drawings and the associated commands. A sheet of double adhesive (or transparency) film can accommodate many units of fluidic channels, with one unit (22 × 61 mm, see supporting information SI) for one layer of a chip (see Appendix A). Note that, each unit has two tags for aligning layers of films. Also, the holes, channels, horizontal lines, and vertical lines should be clustered onto separated Adobe Illustrator layers, with all holes on one layer, horizontal lines on another, and so on (see Appendix A). Such separations allow pauses between cuts of Illustrator layers, and the cutting blade can be cleaned during the breaks. The removal of debris is important to prevent the film from being tangled with the blade. Also, with the above separations, different forces and patterns of cutting strokes can be applied to different Adobe Illustrator layers. Finally, the diameters of holes in double adhesive films should be slightly larger than those in transparency films and coverslips. This is to minimize the contact of fluids with the exposed surface of misaligned double adhesive films. Also, the sizes of the units of double adhesive films and transparency film were the same, yet smaller than that of a coverslip.

For the Lab-Chip-For-Fish, the double adhesive films and transparency films were similarly designed and cut as above (see Appendix A). PMMA plates (60 × 24 × 3 mm, see Appendix A) with large (4 mm diameter) and small (2 mm diameter) holes were made by a local machine shop.

**Use of alignment templates to stack units of transparency and double adhesive films.** The alignment template consists three parts, a film support, a slide frame, and a holder (see Appendix A). The film support is a rectangle transparency film (60 × 100 mm) with marks for positioning one unit double adhesive (or transparency) film. The slide frame is a 1 mm-thick PMMA plate (also 60 × 100 mm) with a central hollow rectangle (26 × 76 mm) to hold the glass slide. The holder is a 4 mm-thick PMMA plate with rectangular walls to tightly hold the film support and slide frame. A unit of double adhesive (or transparency) film was temporarily fixed on the support via the tags and Scotch tapes (see Appendix A). By placing a slide-containing slide frame into the holder where the U1 double adhesive film was already placed, the U1 adhered to the slide. Similarly, U2-transparency film and U3 adhesive films could be sequentially and accurately piled up (see Appendix A). Lastly, a coverslip with holes was placed on the top of U3 to complete the fabrication of a microfluidic chip.

## 3. Results

For our research and teaching needs, we developed the following fabrications of chip and port to immobilized protein in specific area of the chip.

### 3.1. Lab-on-Chip Immobilization of GFP Protein

The immobilization of proteins in the surface of microfluidic chips is a frequently performed task in the applications of Lab-on-Chip technology (e.g., [31,32,33,34]). 

#### 3.1.1. Design and Make of Chip Port

The port (DK1 port) is made of a block of Teflon (polytetra-fluoroethylene, PTFE) (Figure 1a). The bottom side of the port contains two openings for the input and output of fluid flows (Figure 1b). The left bottom opening leads to a mini three-way switch which is made by drilling a hollow cylinder onto the front wall of the port and fitting the cylinder with a rod having T-shaped channels (Figure 1b). 

Also, on the top and left wall of the port, two tunnels are drilled to match the channels of the rod (Figure 1b). The rod is also drilled to hold a cap screw as the handle of the switch. Moreover, the opening on the top of the port is slightly enlarged to form a sample reservoir for the manual deposit of sample fluid by a pipette (Figure 1b). The reservoir and vertical channel of DK1 port can accommodate 9 μL of sample. The right bottom opening is with a vertical passage which forms two horizontal yet orthogonal branches to the front and right walls of the port (Figure 1c,d). The three-way switch can be set for the ‘top-only’ entry (Figure 1e, upper panel) or the ‘side-only’ entry (Figure 1e, lower panel) of fluid. The sizes of the holes on the side walls allow snap-plug of 2.4-mm tube for the delivery of fluids.

#### 3.1.2. Design and Fabrication of Microfluidic Chip

The chip is made by laying units of Craft Robo-treated double adhesive films, transparency films, and a coverslip over a glass slide (Figure 2a). Figure 2b shows that the chip contains openings for the in and out of fluid, bubble traps, and a protein chamber where GFP protein will be immobilized. In addition, the chip also has three closed minicells for loading of control samples. To precisely align the layers, alignment templates was used (see supporting information (SI) for detail).

We used a craft cutter to incise microfluidic channels in double adhesive films (Figure 3a). The cutter was also used to carve holes on transparency film. To drill holes on coverslips, we used a mini electrical grinder and a diamond coated burr-tip (Figure 3a). The coverslip was placed between a paper pad and a guide plate with the same size of the coverslip (Figure 3b). The guide plate has holes for the grinder to work downward. The use of a guide plate also avoids fingerprints being left on the coverslip. Although a diamond pen can also be used to cut holes on a coverslip, the use a mini grinder is much easier.

#### 3.1.3. Assembly of Port and Chip

The microfluidic chip was completed by attaching the port over the bare microfluidic chip with a piece of double adhesive film having holes (U4-2X, Figure 2a). A properly assembled device lasted at least three weeks without leakage of fluid flow. The port can be removed from the chip with twists and pulls, and cleaned for reuse.

#### 3.1.4. On-Chip Immobilization of GFP Protein

We used the chemistry of Schiff base formation to covalently immobilize protein on the coverslip (Figure 4a). Prior to being assembled into the microfluidic chip, the surface of coverslip was functionalized with amino groups. The protocol of protein immobilization was listed in Table 1. Briefly, a PBS (phosphate buffer saline pH 7.4) solution was delivered to the chip via gate 4 (Figure 4b) and flow out through other gates, individually and sequentially. To functionalize the coverslip with terminal aldehyde groups, a glutaraldehyde solution was added to the chip via gate 1 and out via gate 3. After disconnecting the syringe at gate 1, the channel was flushed with PBS to remove excess glutaraldehyde. Then a 16-μL aliquot of GFP protein was manually loaded via gate 2 while a gentle suction was applied by withdrawing the syringe at port 3 (Figure 4b). Finally, PBS solution was introduced via gate 4 to remove the unbound GFP, and the wash flowed out via gate 1. The flow rates of flushing the microfluidic channels of this unit generally were 0.5 or 1.0 mL/min.

The immobilization of GFP protein on the coverslip was evident as judged from the widespread green fluorescence (Figure 4c). The intensity of the immobilized GFP was ~50–240 fmole/mm^2^, as visually compared with the fluorescence of the standard cells (Figure 4d). However, non-specific binding of GFP protein occurred on the rims of double adhesive films and transparency film.

### 3.2. Lab-Chip-For-Fish for Studying the Development of Fish Embryos

Microfluidic chips have been used to study the developments of embryos and the growth of cells (e.g., [35,36,37]). Based on the above experiences and by incorporating the concepts of Lego, we developed an approach of building versatile microfluidic chips to study the developments of fish embryos (Figure 5). 

This system contains a port, chip, and bridge. First, we made a new port. Although the DK1 port worked fine for the protein chip described above, it had two limits. Occasionally, tubes fell off from the port in a prolonged usage (>12 h). Also, we found that the plug-and-go tube connection did not support the tubes with diameter larger 3 mm due to the shallowness of the plug. Accordingly, we designed the DK2 port, which also had four gates. The DK2 port (Figure 5a) was made of four tubing connectors and a block of PTFE. Second, the fish chip consisted of units of coverslips, double adhesive films, and PMMA plate (Figure 5a). The PMMA plate had large holes as fish cells and small holes as bubble traps. Third, we made a bridge with tunnels to channel fluid flows between the port and the chip (Figure 5a,b). Double adhesive films were used to bind the bridge to the port and the chip. With a bridge, the port does not have to be directly fixed over the chip. Also, two fish chips could be assembled together to form a small array (Figure 5b). A DK2 port channeled fluid flows from one chip to next one (Figure 5c).

The fish chips were used to study the developments of individual embryos of zebrafish (Figure 6). The embryos of 5 hpf (hours post fertilization) were placed in the cells, and the developments of embryos were periodically photographed with a home scanner (Figure 6). The larva fishes could be retrieved from the chips at the end of observation (see materials and methods). The chips were scanned with three methods, reflection without a top cover (Figure 6a), reflection with a black cover (Figure 6b,d), and transmission (Figure 6c,e,f). Apparently, the presented system of chip array supported the developments of fish embryos. Also, an image scanned with transmitting light was best in revealing the main characteristics of embryos and baby fishes. Due to the color of the background, the features of embryos were not distinct in images scanned by the reflected light without top cover (Figure 6a).

## 4. Discussion

The use of commonly available materials and equipment are necessary for cost-effective and straightforward fabrication of microfluidic devices. The mini three-way switch of DK1 port represents a unique design for loading sample and buffer. The uses of traditional syringe and tube to push sample into microfluidic channels are associated with a large dead volume. With the DK1 port, samples of limited volume (5~15 μL) can be conveniently loaded into the reservoir by using a pipette and simultaneously drawn into the microfluidic channel by a pump-generated pull from the output end. Moreover, the switch can then be turned from the ‘top-entry’ mode into the ‘side-entry’ one to irrigate the channel with a large amount of buffer. Researchers had made open ports for manually loading of the sample only into empty microfluidic channels [17]. However, in most cases, the microfluidic channels will be sequentially filled with different samples and buffers. Due to the strong retention of fluid in microfluidic channels, it will be difficult to use a pipette and manually push the fluidic sample through a horizontal microfluidic channel which has been filled with liquid. The above difficulty can be solved by using the mini three-way switch of DK1 port.

The designs of DK1 and DK2 ports are to simplify the tasks of users and manufacturers. Users just plug tubes into the holes of DK1 port and no other treatment (e.g., glue) needed. While tubes many fall off from DK1 port, DK2 port is advantageous by providing secured connections. Inspired by the connectors of electronic devices, researchers developed several connectors, such as the world-to-chip socket [14,38], fit-to-flow connector [19,39], D-subminiature connector [20], force fit connector [24], and plug-socket [21]. Those connectors are also convenient to use but need sophisticated processes to make. On the other hand, the fabrications of DK1 and DK2 ports are relatively straightforward and inexpensive.

The use of a mini-grinder to drill holes on coverslips opens a novel avenue of fabricating microfluidic chips using coverslips. The thinness, transparency, and well-established chemistry of surface modifications make coverslips an ideal material for fabricating microfluidic chips. Also, coverslips are resistant to strong chemicals, while plastic films are labile. However, for a non-industrial grade laboratory, it is difficult to drill (or cut) small holes on coverslips. Holes can be custom-made on coverslips at high prices. Also, lithography techniques have to be used to etch channels on glass substrates. Therefore, not many microfluidic chips are made of glass. With the brilliant use of a mini-grinder, we demonstrated that I/O holes of microfluidic chips could be made on coverslips. Since a mini-grinder is inexpensive (< $30) and easy to handle, it is worth advocating for the use of this tool.

The extensive uses of double adhesive films and transparency films are noteworthy for the following reasons. First, as reported earlier [9], an obvious advantage is in the cost-effectiveness of using those films and a Craft Robot. Second, it takes little time and labor to fabricate microfluidic chips. In addition to this study, other researchers have used these films alone or with other materials to construct interesting microfluidic chips [6,8,10,11,12,40,41]. Third. the binding by double adhesive films is durable, yet not permanent. The M60 double adhesive film can resist a pulling force up to 1.2 kg/inch^2^. In our experiences, as long as the channels and holes were swathed by double adhesive film of at least 3 mm width, the fish chip can bear a 5 mL/h fluid flow for at least three days. Without being disturbed by external forces, the binding can last at least three weeks. At the end of experiments, the ports can be removed by twisting and pulling, and then cleaned for reuses. Compared with other re-usable ports which are bound to chips via screws (e.g., [24,25,26]), manual press fit (e.g., [19,20,22,25]), vacuum press (e.g., [19,28]), clamp (e.g., [38]), or magnet (e.g., [18,23]), the use of double adhesive films is equally simple and effective. However, in terms of the processes and cost of fabricating those ports and chips, the use of double adhesive films probably is the simplest solution. Finally, the DK ports can also be used with microfluidic chips made of PDMS. A piece of double adhesive film can be used to bind DK port with PDMS. Alternatively, they can be connected by a bridge and double adhesive films.

The use of double adhesive films and transparency films also brings in the invention of the bridge. With a bridge, the port does not have to be stuck on the chip. There is no need of fabricating a new port for a new chip. By making a new bridge, an old port and be used with a new chip. Moreover, the observation of experimental results will be easier without connecting tubes on the chip.

The chip array is another novelty. As demonstrated by our Lab-Chip-For-Fish array, individual chips can be assembled with a bridge to form an array. Alternatively, the PMMA blocks with holes can be assembled and sandwiched between layers of Craft Robo-treated double adhesive films and transparency films. Since various routes of liquid flows can be designed to go through the holes of PMMA blocks, numerous chip arrays can be created. The idea of building chip arrays has been proposed recently by other teams, e.g., a reconfigurable stick-n-play modular microfluidic system built from magnetic interconnections [18]. However, our use of double adhesive films and transparency films to create a chip array is advantageous in terms of the cost and labor.

The main shortcoming of using double adhesive films and transparency films is that these materials are labile in many organic solvents. Accordingly, users need to test the chemical compatibility of their film materials beforehand.

Regarding to the cost of first-time building the presented devices, the Craft-Robo cost ~200 US dollars ($), the mini grinder ~30$, and the letter-size double adhesive film <1$/sheet. Teflon blocks were retrieved from the institute warehouse without further costs, and PMMA plates were bought at 5$/ft^2^. The labor cost for out-sourced machining each DK1 or DK2 port was 100$, and 50$ for working on PMMA plates. The labor cost can be waived if one can DIY in the lab. Since the craft cutter, mini grinder, the DK ports and PMMA chips are re-usable and the consumables (transparency films, double-adhesive films, coverslips and slides) are purchased at enough quantities, the later-on device building needs no further costs.

Despite of the interesting features of building the presented model units, there are limits. Firstly, the double adhesive films, transparency films and plastic blocks may be labile to certain organic solvents and thus, users should test the chemical compatibility of their materials beforehand. Double adhesive films and transparency films of various materials and properties are available for different applications. Secondly, double-adhesive films have finite capacity of bonding substrates. For example, the M60 double adhesive film can resist a pulling force up to 1.2 kg/inch^2^. Leakages of fluids will happen when the channel pressure by high flow rates exceeds the binding capacity of the double adhesive film. We have no information about the upper end of permissive flow rates for our devices, due to the lack of equipment. Nevertheless, the flow rates of 1 mL/h ~1 mL/min have been applied to both model units without problem.

## 5. Conclusions

Re-usability and modularization are the trends of chip-to-world connectivity. The adoption of a re-useable port (or connector) saves multiple costs, time, and labor. A port module is necessary for versatile designs of microfluidic devices and friendly user-experiences (e.g., plug-n-play). In addition, a port with the switches of sample loading by pipet or by a syringe is very desirable. However, the needs of sophisticated equipment will deter interested yet not-well funded researchers. We demonstrated that microfluidic devices could be made with simple and inexpensive materials and tools.

## Figures and Tables

**Figure 1 sensors-18-04223-f001:**
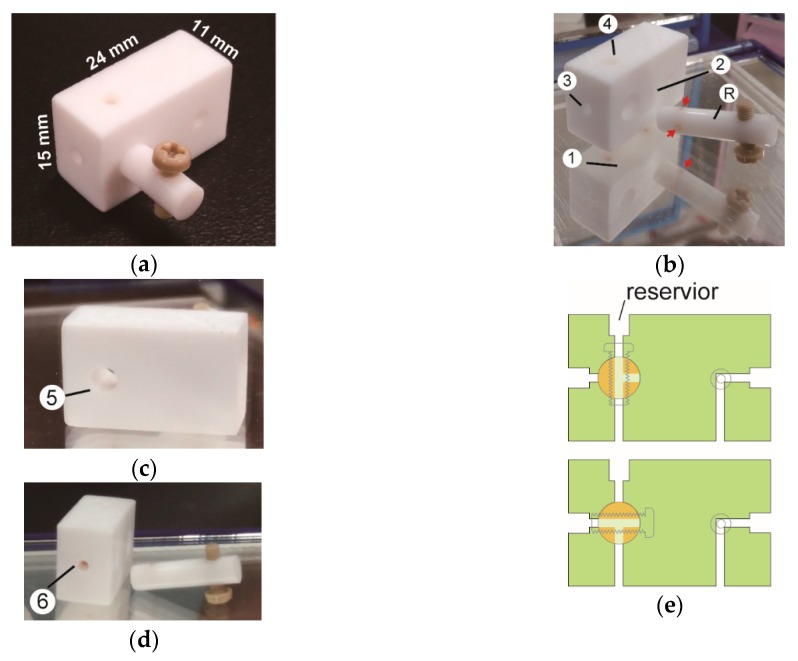
DK1 port for microfluidic protein chip. (**a**) The port is equipped with a three-way switch. (**b**) The bottom (①), front (②), left side (③), and top (④) views of a disassembled DK port with the rod (Ⓡ) of the three-way switch. Three red arrows indicate the openings of T-shaped tunnels. (**c**) and (**d**) The back (⑤) and right side (⑥) views of the disassembled DK port. Mirrors were placed below and around the parts to show all sides of the port. (**e**) The cross-section views of the DK1 port showing the positions of the three-way switch for the ‘top-only’ (upper panel) and the ‘side-only’ (lower panel) entries of fluids.

**Figure 2 sensors-18-04223-f002:**
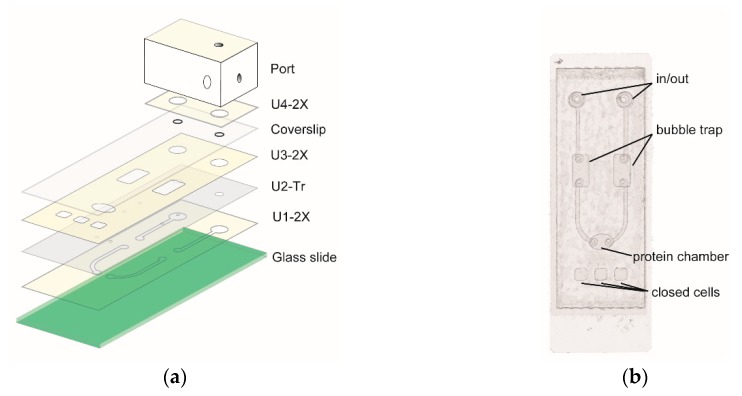
Microfluidic protein chip. (**a**) Scheme of laying units of double adhesive films (2X), transparency film (Tr), coverslip, and port on the glass slide. (**b**) An assembled chip (top view).

**Figure 3 sensors-18-04223-f003:**
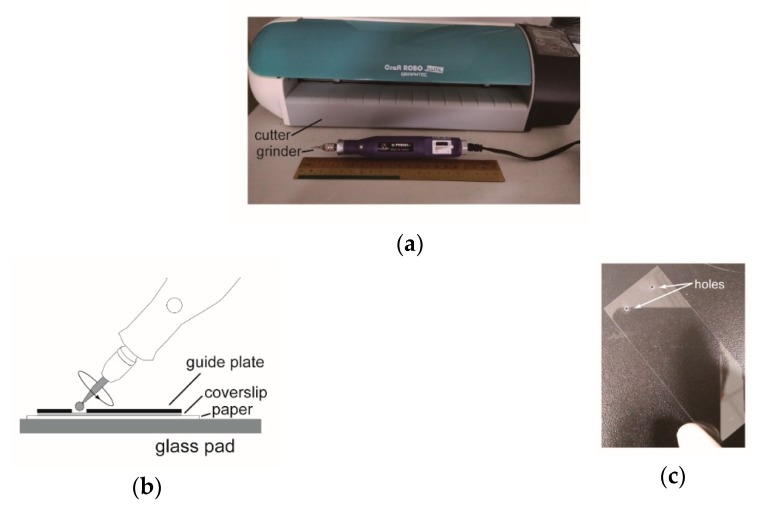
Tools used in making microfluidic chips. (**a**) A craft paper cutter and mini grinder. A 20-cm ruler is also displayed. (**b**) The use of a mini grinder to drill holes on a coverslip. (**c**) A coverslip with two holes drilled by the mini grinder.

**Figure 4 sensors-18-04223-f004:**
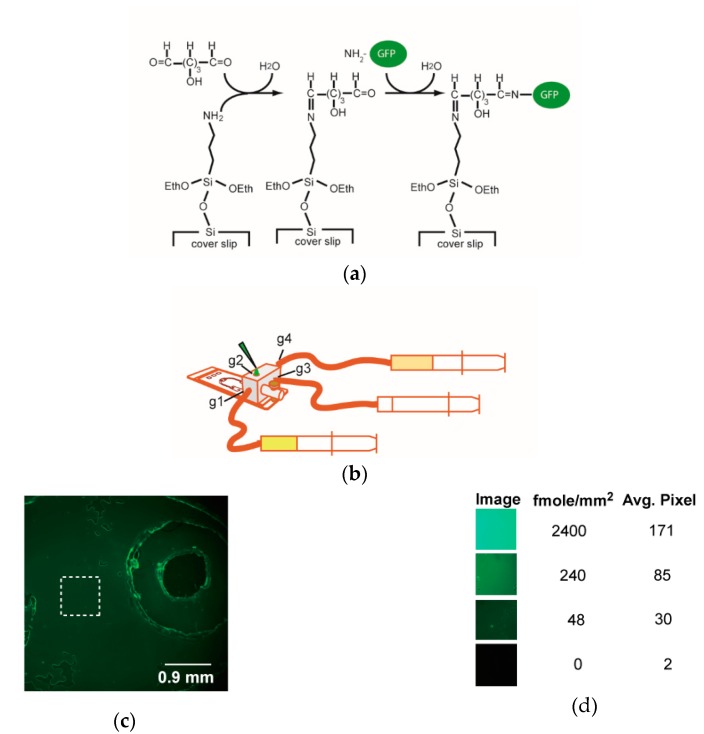
GFP immobilization on the microfluidic chip. (**a**) The chemistry of immobilizing GFP protein on coverslip. Glutaraldehyde reacts with the amino groups of surface silane and GFP protein. (**b**) The setup of microfluidic chip and peripherals for immobilizing GFP. g1 represents gate 1, similarly for g2~g4. (**c**) GFP protein immobilized as revealed by the green fluorescence within the region of interest. The average density of GFP protein within the dash line square is about 50 fmole/mm^2^ as compared with those of the standards. (**d**) Semi-quantitation of GFP. To estimate the efficiency of protein immobilization, three aliquots (0.8 μL) of ‘standard’ solutions with different concentrations of GFP protein (40, 4, and 0.8 μM) were spotted into three nearby closed cells and sealed in the chip. The fluorescent images of the ‘standard’ cells were taken, and the amount of GFP protein in unit area (fmole/mm^2^) of those cells were estimated. The image between two closed cells was also taken as 0 GFP. The average pixels of the 8-bit images of the standards were measured by using software ImageJ. Two repeats of device-building and GFP-immobilization yielded similar results.

**Figure 5 sensors-18-04223-f005:**
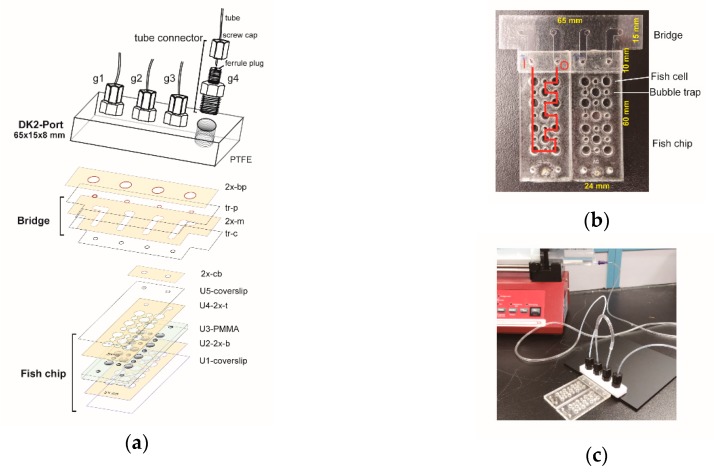
Fish chip with Lego concept. (**a**) Schematic diagram showing the components and assembly of DK2 port, bridge, and fish chip. Double adhesive films were cut with a computer-controlled cutter to form U2-2x-b, U4-2x-t, 2x-m, 2x-cb. 2x-m, and 2x-bp. Transparency films were similarly cut to form tr-c and tr-p. Four gates (g1~g4) were formed by attaching tube connectors to PTFE. Tubes were inserted into the ferrule plugs and firmly attached to the gates by tightening screw caps. (**b**) A bridge with two fish chips. The red line indicates the input (I) and output (O) path of fluid flow in the chip channels. (**c**) A functional chip array. A syringe pump continuously injected fresh normal phosphate buffer saline to the fish cells where fish embryos had been placed. The rate of fluid flow was 5 mL/h.

**Figure 6 sensors-18-04223-f006:**
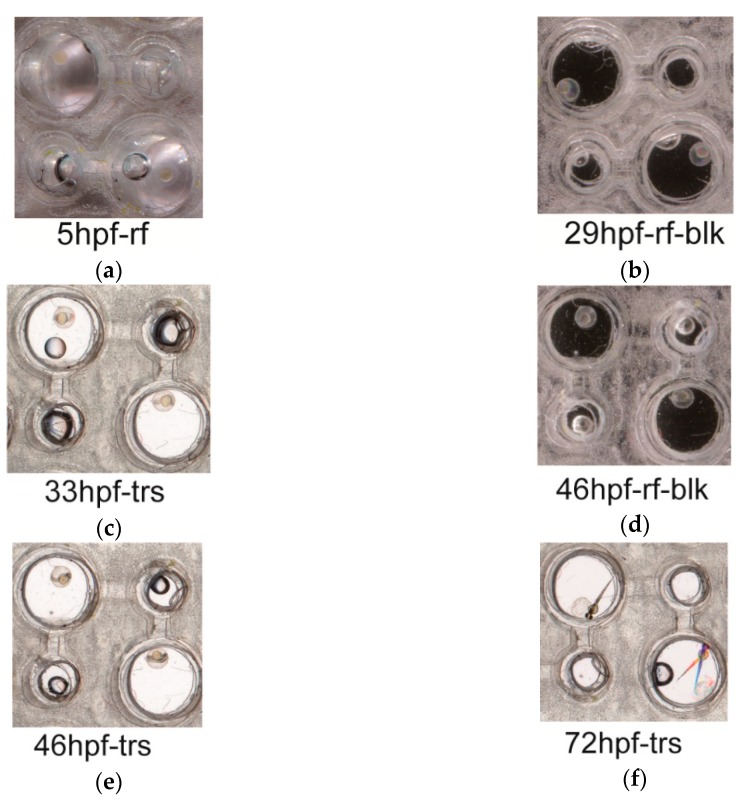
The developments of fish embryos in fish cells. Selected embryos of zebra fish were placed in the fish cells, and their developments at 5 hpf (**a**), 29 hpf (**b**), 33 hpf (**c**), 46 hpf (**d,e**), and 72 hpf (**f**) were photographed at 1200 dpi with a home scanner. Image scans were conducted with transmitting light (**c,e,f**) or reflective light without (**a**) or with a black (**b,d**) top cover. Dual fish images at the lower right cell (**f**) were caused by the movements of the larva fish during the scan. Methods—the fertilized eggs with signs of cell divisions were collected at ~4.5 hpf and placed in the chambers of the chip for first observation. A flow rate of 4 mL/h. was applied to the device. Since the color and transparency of the fish body varied during the embryo development, we tried three different modes of image scanning to take pictures of the fish embryos. Apparently, the scans with the reflective light without a black background did not yield clear images of the 5 hpf embryos. Also, the images by the reflect light with a black background were not as clear as those by the transmitted light. Three repeats of device-building and embryo-cultivation yielded similar results.

**Table 1 sensors-18-04223-t001:** Steps of on-chip immobilizing GFP protein on a coverslip.

Steps	Purposes	Gate1	Gate2	Gate3	Gate4	Notes
1	Fill fluid through all gates	Open or closed	Open or closed	Open or closed	Inject PBS	For the gates 1~3, one gate is open, two are closed
2	To modify the surface of coverslip with glutaraldehyde	Inject 8% glutaraldehyde	Closed	Opened	Closed	Incubate for 30 min
3	Remove excess glutaraldehyde	Open (no syringe)	Closed	Opened	Inject PBS	
4	Load and immobilize GFP	Closed	Manually load GFP	Suction applied	Closed	Incubate for 30 min
5	Remove excess GFP	Open (no syringe)	Closed	Closed	Inject PBS

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
