# Peer review of "Workshop, Cost-Effective and Streamlined Fabrications of Re-Usable World-To-Chip Connectors for Handling Sample of Limited Volume and for Assembling Chip Array"

_sensors, 2018, doi:10.3390/s18124223_

Round 1

Reviewer 1 Report

The manuscript by Lue et al., reports processes for fabricating simple connectors for microfluidics devices. Overall, the manuscript needs to more clearly express what the novelty of the proposed DK1 and DK2 ports are compared to currently available systems. 

Specifically the following points should be addressed:

The term 'fish chip' is not very technical - this should be changed or at least put in inverted commas. 

All figures need scale bars/dimensions including on them.

What volumes can be accommodated using the ports?

What flow rates/pressures are the ports capable of dealing with?  

What are the production costs?

Very limited data is presented from the 'protein chip'. No fluorescent values, replicate data, errors etc. 

Again, little data is presented on the 'fish chip'. No information on fertilisation of embryos, unclear why different backgrounds were used at different time periods, no explanation of what the results mean, no replicate data provided. 

Why have these two examples been chosen compared to any other microfluidic applications?

Final paragraph of results/discussion mentions that the ports are unlikely to work with solvents but offers no solutions. 

Author Response

First of all, we would like to thank the editor for handling our paper and the reviewers for their hard work and constructive comments. We found the reviews very helpful in further improving the quality of our manuscript. The following is a list of item-by-item responses to the review comments.

Reviewer 1:

Q1: The term 'fish chip' is not very technical - this should be changed or at least put in inverted commas.

A1: Thanks for this comment. We have rename it. It is called Lab-Chip-For-Fish.

Q2: All figures need scale bars/dimensions including on them.

A2: Thanks for this comment. We have indicated the sizes of the objects or included scale bars in the figures. Specifically, the size of DK1 port is shown in Figure 1A; the size of the protein chip is described in the legends of Figure 2; in Figure 3, the sizes of the Craft paper cutter and the mini grinder are compared with a 20-cm ruler; in Figure 4C, a 0.9 mm scale bar is placed near the hole of the protein chip; the size of DK2 port is indicated in Figure 5A, and the sizes of the bridge and chip are shown in Figure 5B; and finally in Figure 6, a 4-mm scale bar is drawn to show the diameter of the fish-accommodating holes.

Q3: What volumes can be accommodated using the ports?

A3: Thanks for asking this question. In the LL266-268, p9 of the revised manuscript, we mention that "With the DK1 port, samples of limited volume (5~15 ml) can be conveniently loaded into the reservoir by using a pipette and simultaneously drawn into the microfluidic channel by a pump-generated pull from the output end." We added a statement that "DK1 accommodates 9 ml of sample" in LL161-162, p4 of the revised manuscript. In the LL104-105, p3 of the revised manuscript, "the volume of the channel within the gate of DK2 port is ~28 ml."

Q4: What flow rates/pressures are the ports capable of dealing with? 

A4: Thanks for asking this question. The M60 double adhesive film can resist a pulling force up to 1.2 kg/inch2 (in the revised manuscript, LL71-72, p2). Honestly, we did not investigate this issue, for the lack of equipment. Nevertheless, the flow rates of 1 ml/hr. ~ 1 ml/min have been applied to both model units without problem (in the revised manuscript, LL342-343, p10).

Q5: What are the production costs?

A5: Thanks for asking this question. We now added a paragraph for the asked question (in the revised manuscript, LL326-333, p10): "the CraftRobo costed ~200 US dollars ($), the mini grinder ~30 $, the Letter-size double adhesive film < 1 $ /sheet. Teflon blocks were retrieved from the institute warehouse without further costs, and PMMA plates were bought at 5 $/ft2. The labor cost for out-sourced machining each DK1 or DK2 port was 100 $. The labor cost can be waived if one can DIY in the lab."

Q6: Very limited data is presented from the 'protein chip'. No fluorescent values, replicate data, errors etc.

A6: Thanks for asking these questions. We don't have equipment for the asked tasks. To semi-quantify the amount of GFP immobilized on the coverslips, we added GFP protein of different concentrations in the near-by "standard" cells, and visually compared the fluorescence of the oval area of the chip with that of the standards. We have rephrased these parts in the revised manuscript LL204-209, p7

"Semi-quantitation of GFP. To overcome the lack of quantification equipment, three aliquots (0.8 ml) of "standard" solutions with different concentrations of GFP protein (40, 4 and 0.8 mM) were spotted into three nearby closed cells and sealed in the chip. The fluorescent images of the "standard" cells were taken, and the amount of GFP protein in unit area (fmole/mm2) of those cells were estimated. The image between two closed cells was also taken as 0 GFP. Two repeats of device-building and GFP-immobilization yielded similar results."

and LL211-212, p7.

"The intensity of the immobilized GFP was ~ 50 – 240 fmole/mm2, as visually compared with the fluorescence of the standard cells (Fig. 4D)"

Q7: Again, little data is presented on the 'fish chip'. No information on fertilisation of embryos, unclear why different backgrounds were used at different time periods, no explanation of what the results mean, no replicate data provided.

A7: Thanks for asking these questions. Since this report is about building devices with the proposed materials and methods, we did not want to put too much detail about the cultivation of fish embryos. Nevertheless, we added a short paragraph in the legends of Figure 6 (LL255-261, p9 of the revised manuscript) to address the reviewer's questions.

"Methods, the fertilized eggs with signs of cell divisions were collected at ~ 4.5 HPF and placed in the chambers of the chip for first observation. A flow rate of 4 ml/hr. was applied to the device. Since the color and transparency of the fish body varied during the embryo development, we tried 3 different modes of image scanning to take pictures of the fish embryos. Apparently, the scans with the reflective light without a black background didn't yield clear images of the 5 HPF embryos. Also, the images by the reflect light with a black background were not as clear as those by the transmitted light. Three repeats of device-building and embryo-cultivation yielded similar results."

Q8: Why have these two examples been chosen compared to any other microfluidic applications?

A8: We chose these two applications among numerous others, just because in our teaching and research we need to immobilize protein in microfluidic channels or study the developments of fish embryos in various conditions.

Q9: Final paragraph of results/discussion mentions that the ports are unlikely to work with solvents but offers no solutions.

A9: Thanks for the comments. We have reminded readers that films made of different materials and properties are available (in the revised manuscript LL326-328, p10).

"Regarding to the cost of first-time building the presented devices, the Craft-Robo costed ~200 US dollars ($), the mini grinder ~30 $, and the Letter-size double adhesive film < 1 $ /sheet. Teflon blocks were retrieved from the institute warehouse without further costs, and PMMA plates were bought at 5 $/ft2."

Reviewer 2 Report

This work describes a methodology of how to construct a chip-to-world interface made of Teflon, which provides multiple features for fluid handling. In addition, due to the lack of equipment for soft lithography, the authors assembled microfluidic chips by placing parallel double adhesive film between a glass slide and coverslip. However, even though the topic of this study is of interest, the idea is not original as evidenced by recently published papers dealing with related topics such as connecting microfluidic chips using a chemically inert, reversible, multichannel chip-to world ( Lab on a chip, 2013, 13, 4343–51) or a chip-to-world connector with a built-in reservoir for simple small-volume sample injection (Lab on a chip, 2014, 14, 178–81). In general, the manuscript is written in a clear way, being easily understandable with experimental results on various applications proving the design's performance. I strongly believe that the described methodology represents a highly valuable platform for researchers interested in the technological transfer of lab-on-a-chip platforms from complex and sophisticated fabrication laboratories to non-specialized biological labs to customize microfluidic experimentation and will as such capture the attention of a broad scientific community. Thus, I would recommend publication of this manuscript.

Author Response

First of all, we would like to thank the editor for handling our paper and the reviewers for their hard work and constructive comments. We found the reviews very helpful in further improving the quality of our manuscript. The following is a list of item-by-item responses to the review comments.

Reviewer 2:

1: This work describes a methodology of how to construct a chip-to-world interface made of Teflon, which provides multiple features for fluid handling. In addition, due to the lack of equipment for soft lithography, the authors assembled microfluidic chips by placing parallel double adhesive film between a glass slide and coverslip. However, even though the topic of this study is of interest, the idea is not original as evidenced by recently published papers dealing with related topics such as connecting microfluidic chips using a chemically inert, reversible, multichannel chip-to world ( Lab on a chip, 2013, 13, 4343–51) or a chip-to-world connector with a built-in reservoir for simple small-volume sample injection (Lab on a chip, 2014, 14, 178–81). In general, the manuscript is written in a clear way, being easily understandable with experimental results on various applications proving the design's performance. I strongly believe that the described methodology represents a highly valuable platform for researchers interested in the technological transfer of lab-on-a-chip platforms from complex and sophisticated fabrication laboratories to non-specialized biological labs to customize microfluidic experimentation and will as such capture the attention of a broad scientific community. Thus, I would recommend publication of this manuscript.

A1: We appreciated the comments. Thanks.

Round 2

Reviewer 1 Report

Overall, the authors have been very attentive to the comments made on the manuscript and have made important improvements. I would still argue that the fluorescent data could be easily analysed using a freely available software such as Image J to enable more precise measurements to be made allowing intensity to be measured against the concentrations and plotting of replicates. Otherwise I believe the manuscript should be accepted for publication, with the above suggestion addressed. 

Author Response

First of all, we would like to thank the editor for handling our paper and the reviewers for their hard work and constructive comments. We found the reviews very helpful in further improving the quality of our manuscript. The following is a list of item-by-item responses to the review comments.

Reviewer 1:

Q1: Overall, the authors have been very attentive to the comments made on the manuscript and have made important improvements. I would still argue that the fluorescent data could be easily analysed using a freely available software such as Image J to enable more precise measurements to be made allowing intensity to be measured against the concentrations and plotting of replicates. Otherwise I believe the manuscript should be accepted for publication, with the above suggestion addressed.

A1: Thanks for this comment. By using ImageJ software, we have measured the averaged pixels of the 8-bit images of the standards (Figure 4d, revised manuscript). We also used those data to estimate the average density of GFP protein in a specific area of our protein chip, as shown in the Figure 4c of the revised manuscript. The changes of figure legends are highlighted.
